# Overview of Transcriptomic Research on Type 2 Diabetes: Challenges and Perspectives

**DOI:** 10.3390/genes13071176

**Published:** 2022-06-30

**Authors:** Ziravard N. Tonyan, Yulia A. Nasykhova, Maria M. Danilova, Yury A. Barbitoff, Anton I. Changalidi, Anastasiia A. Mikhailova, Andrey S. Glotov

**Affiliations:** 1Department of Genomic Medicine, D.O. Ott Research Institute of Obstetrics, Gynaecology and Reproductology, 199034 Saint-Petersburg, Russia; ziravard@yandex.ru (Z.N.T.); yulnasa@gmail.com (Y.A.N.); elenamariamassa@gmail.com (M.M.D.); barbitoff@bk.ru (Y.A.B.); anton@bioinf.me (A.I.C.); anamikhajlova@gmail.com (A.A.M.); 2Bioinformatics Institute, 197342 Saint-Petersburg, Russia; 3Department of Genetics and Biotechnology, St. Petersburg State University, 199034 Saint-Petersburg, Russia; 4Faculty of Software Engineering and Computer Systems, ITMO University, 197101 Saint-Petersburg, Russia

**Keywords:** type 2 diabetes, transcriptome, RNA-seq, microarray, single-cell, gene expression

## Abstract

Type 2 diabetes (T2D) is a common chronic disease whose etiology is known to have a strong genetic component. Standard genetic approaches, although allowing for the detection of a number of gene variants associated with the disease as well as differentially expressed genes, cannot fully explain the hereditary factor in T2D. The explosive growth in the genomic sequencing technologies over the last decades provided an exceptional impetus for transcriptomic studies and new approaches to gene expression measurement, such as RNA-sequencing (RNA-seq) and single-cell technologies. The transcriptomic analysis has the potential to find new biomarkers to identify risk groups for developing T2D and its microvascular and macrovascular complications, which will significantly affect the strategies for early diagnosis, treatment, and preventing the development of complications. In this article, we focused on transcriptomic studies conducted using expression arrays, RNA-seq, and single-cell sequencing to highlight recent findings related to T2D and challenges associated with transcriptome experiments.

## 1. Introduction

T2D mellitus is a chronic metabolic disease affecting 6.28% of the world’s population and characterized by β-cell dysfunction and persistent hyperglycemia [1,2]. It can lead to many microvascular and macrovascular complications, including kidney failure, diabetic retinopathy, peripheral neuropathy, heart attack, and stroke, which are associated with a worse quality of life in affected people [3,4]. T2D currently has a prevalence rate of 6059 cases per 100,000 and is projected to affect 7079 individuals per 100,000 by 2030. The low quality of life of affected people and the growing incidence of T2D around the world, especially among young people, make T2D one of the main concerns of public health worldwide [2]. A good understanding of the etiological factors and pathogenetic mechanisms is needed to identify risk factors and prevent the development of this disease or delay the onset of its severe complications. The risk assessment of T2D requires the consideration of not only BMI, age, and environmental factors, such as food intake and physical activity, but also inheritance [5]. The candidate-gene study and Genome-Wide Association Study (GWAS) approaches were used to identify a number of loci and genes associated with the risk of T2D and its complicated course [6,7]. The subsequent study of these new loci revealed new molecular mechanisms underlying the disease. Most of them were responsible for insulin sensitivity, β-cell function, and inflammation. However, the found genes explain only about 10% of the heritability of T2D [8,9]. It becomes clear that single polymorphisms cannot explain the pathogenesis of a complex disease and act as an independent risk factor. The main difficulty lies in finding the link between the found variants and phenotypic manifestations, especially if the gene is not directly involved in glycemic regulation. Moreover, variants of low minor allele frequency that could make a significant contribution to the development of T2D may not be captured by GWAS, which makes the problem of “missing heritability” relevant [10]. Most of the loci identified are located in non-coding regions and play a regulatory role in the control of gene expression [11]. The main purpose of such studies is the development of risk assessment tools for the prediction of T2D and its complications, including the construction of the polygenic risk scores, both based solely on T2D-associated SNPs and adjusted for gender, age, anthropometric measurements, and laboratory parameters. Accounting for numerous trait-associated alleles and non-genetic factors made it possible to increase the predictive power of the models, with the relative performance estimated as areas under receiver operating characteristic curves, whose values increased from 0.571 to 0.901 [12]. However, there are several limitations that might impede the implementation of these instruments in practice. Most of the studies were performed on European populations, and large-scale studies covering people with T2D worldwide are needed due to possible ethnic differences. Another challenge to the application of the polygenic risk score is the interpretation of the results, which requires the consideration of non-genetic risk factors in patients [13]. Moreover, these factors (BMI, age, family history, diet, physical activity), in combination with the laboratory markers of glycemia, can be crucial for determining the individual risk of T2D [14]. Nonetheless, the study of gene expression and epigenetic modifications in people with T2D, coupled with the identification of the rare genetic variants using the approaches of whole genome, whole exome, or target genes sequencing in them, have the potential to fill the gap between the genotype data and the phenotype in diabetic individuals, which will eventually allow for an understanding of genomic function [15]. Therefore, in the last decade, much attention has been paid to studies on transcriptome analysis in T2D patients (Table 1). Transcriptome studies analyze the sum of all RNA transcripts present in an organism. Currently, three main methods are in use for transcriptome analysis: microarray, RNA sequencing, and single cell sequencing. We reviewed the main advantages and shortcomings and also discussed the most promising findings of them and the challenges researchers face when choosing these approaches.

## 2. Approaches for Assessing Gene Expression in T2D

To evaluate gene expression, several methods are widely used in molecular biology, including both conventional (northern blot, RT-qPCR, expression arrays) and more advanced (RNA-seq, single-cell sequencing) technologies (Figure 1).

Northern blot, developed in 1977, is one of the first methods used by researchers to analyze gene expression in T2D patients [46]. It includes the separation of RNA by gel electrophoresis and subsequent hybridization with a probe labeled with chemiluminescent [47]. Even though northern blot is quite cheap and easy to perform and has a high specificity, its sensitivity is lower compared to that of RT-qPCR [48]. Compared to expression arrays, the northern blot is inferior in terms of the number of simultaneously analyzed genes, but its sensitivity is higher [49]. Today, the northern blot is not widely used in the clinic due to the relatively low sensitivity and slow analysis; its use for scientific purposes is also limited [50].

Among the variety of methods applied over time to analyze the expression of genes associated with T2D, quantitative reverse transcription polymerase chain reaction (RT-qPCR), DNA microarrays, and RNA-seq are the most preferred. RT-qPCR is a reliable and widely used technique for analyzing the expression level of target sequences. It is highly sensitive and specific, relatively cheap, allowing for the analysis of large samples, and it does not require a long and multi-stage sample preparation [51]. One of the difficulties in conducting an RT-qPCR experiment is the choice of a reference gene that is expressed relatively constant in tissues, since the accuracy of the results depends on it [52]. RT-qPCR provides reliable data and is used to validate the results of NGS or microarray experiments. It should be noted that the RT-qPCR approach application for expression analysis makes no allowance for the detection of new transcripts, and it also requires hypotheses, pre-designed probes, and, therefore, thorough experiment planning.

A gene expression array (or DNA microarray) is a technology that has been successfully used for measuring and analyzing gene expression for over a decade. The principle of the approach is to hybridize the fluorescently labeled complementary DNA from the control and experimental samples to oligonucleotides attached to a solid surface of the array [53]. The crucial limitation of this approach arises from microarray probes design based on the reference genome. Low binding specificity, leading to cross-hybridization and background signals, may substantially affect the results of the study [54]. The repetitive or related sequences, alternatively spliced transcripts, and SNPs cannot be analyzed using DNA microarray. In addition, the variety of microarrays used by researchers while conducting experiments reduces the reproducibility of the data obtained [55]. Despite the shortcomings, gene expression arrays are still in demand, since microarray technology is cheaper than NGS, and the sample preparation process is simpler and more “traditional”, which allows for the analysis of a relatively larger number of samples. The main stages of the analysis of data obtained using expression arrays include quality assessment and normalization. The main challenge in the analysis of microarray data is to find patterns of differential expression that are able to characterize the difference between samples. Today, two main approaches are used: the first is to identify individual DEGs; the second involves the identification of functionally related gene sets with an altered expression. There are many commercial tools and open-source software packages for finding DEGs in microarray experiments [56]. The main tool for the second approach is Gene Set Enrichment Analysis (GSEA) software. Overrepresented functions are revealed when comparing DEGs against the rest of the genome or when performing a GSEA that assigns ranks to each gene in the transcriptome depending on the differential expression level [57].

Over the past few years, RNA-seq has become a standard approach for gene expression measurement. Initially, the Sanger sequencing method was used for cDNA sequencing, but it was soon replaced by next generation sequencing (NGS). In addition, the Sanger sequencing method was not suitable for the analysis of transcript expression [58]. One of the main limitations of RNA-seq is the high cost, as well as the long and complex sample preparation process, which reduces the sample size. Another difficulty in conducting such an analysis is the lack of optimized standard protocols, as well as the large output file sizes [59].

However, the benefits of this method greatly outweigh its disadvantages. For instance, unlike RT-qPCR and expression arrays, RNA-seq does not require a priori sequence information, which allows for the identification of new splice-variants, SNPs, novel transcripts, and non-coding RNAs [60]. The first stage of RNA-seq results analysis is quality control, including read trimming (the removal of adapter sequences, reads with poor quality, and duplicated reads) and the analysis of sequence quality and GC content [61]. The next step is read alignment to the genome (in case novel transcript discovery is required) or annotated transcriptome (when novel transcript discovery is not needed) and subsequent read counting using various tools for within-sample normalization [62]. Several software tools are available today to analyze DEGs in RNA-seq experiments (DeSeq2, SAMseq, NOISeq, EBSeq, Sleuth, baySeq, edgeR, limma-vomm, etc.). The choice of method mostly depends on the goals, the sample size, and the number of replicates. When comparing the DEGs software tools, the NOIseq, DESeq2, and limma+vomm methods present the best levels of specificity. However, an optimal method for determining DEGs when conducting an RNA-seq experiment is still lacking, so it is recommended to combine different tools to obtain more reliable results [63]. The final step is to assess the biological functions of DEGs as well as the molecular pathways in which they are involved, using the GSEA software [62].

While expression arrays and RNA-seq make it possible to analyze an incomparably larger number of genes, the results of experiments conducted using these methods require validation by RT-qPCR. 

Single-cell sequencing technology became widespread in recent years. This method has a number of advantages over standard RNA-seq: it allows for the study of cell population structures, the identification of the functional and developmental heterogeneity of the cell population, and the distinguishing of a small number of cells [64]. The single-cell phenotyping of transcriptome changes has the potential to detect the expression of cell-specific genes and possibly identify new cell subtypes. The search for DEGs during single-cell experiments is challenging because the data obtained differ significantly from the bulk RNA-seq data due to high heterogeneity, small total read counts, and a large number of zero counts [62,65]. Usually, single-cell libraries are aligned to the reference transcriptome. Both standard software for bulk RNA-seq (DESeq2, edgeR) and special tools for single-cell sequencing (D3E, DEsingle, MAST, Monocle2, SigEM, SCDE, scDD, SINCERA) are used for the analysis. A comparison of these methods showed that special tools do not significantly outperform the standard ones. All the tools have their own advantages and limitations, but none of them meet today’s requirements for solving single-cell sequencing problems, so the search for a precise analysis method continues to this day [66].

## 3. Transcriptome Studies in T2D

The pathogenesis of T2D is not completely understood. The significance of immune inflammation and lipid metabolism in the development of T2D [67] has been described repeatedly; however, transcriptomic studies allow for the supplementation of the available information on potential pathways that are involved in the pathogenesis of diabetes or are affected in T2D patients—for example, the ubiquitin-proteasome system, cell-cycle pathways, and cancer signaling. Appendix A presents the main DEGs and biological processes identified by transcriptome studies.

### 3.1. DEGs Involved in Lipid Metabolism

Adipose tissue plays a significant role in the development of insulin resistance in T2D [68]. According to modern concepts, the role of adipose tissue in the pathogenesis of diabetes comes down to the accumulation of excess lipids in the liver and to inflammation accompanied by inflammatory cytokine production, leading to impaired insulin signaling [69]. A recent RNA-seq study in visceral adipose tissue showed the reduced expression of genes responsible for fatty acid synthesis and mitochondrial function and the increased expression of genes associated with innate immunity and transcriptional regulation in obese diabetic women [39]. While this study was mainly focused on visceral adipose tissue, Saxena and colleagues directed their attention to peripheral subcutaneous adipose tissue. The researchers compared expression profiles in thigh subcutaneous adipose tissue cells in 30 diabetic patients and 30 controls. A total of 971 differentially expressed genes (DEGs) were found. Pathway enrichment analysis revealed immune and inflammatory response signaling pathways, including the HIF-1 signaling pathway, activated in the obesity-induced chronic hypoxic state and involved in the regulation of proinflammatory cytokines, endothelial NO synthase, insulin signaling, and gluconeogenesis [70,71]. The upregulation of elements of the inflammatory NF-kappa B signaling pathway was also determined, which allowed for the conclusion that subcutaneous adipose tissue was inflamed in T2D patients and visceral fat along with the activation of other inflammation-related pathways. Additionally, pathways were found to be directly involved in fat metabolism (AMPK signaling pathway, Adipocytokine signaling pathway) and glucose metabolism (insulin signaling pathway, PI3K-Akt pathway) [27].

Saxena and colleagues repeated the comparison of the expression profiles of femoral fat in T2D patients and healthy controls using RNA-seq and found an altered expression for 2752 genes and 65 miRNAs. The expression of 10 genes was validated by qPCR (*AKT2*, *BCL6*, *LIPE*, *METTL9*, *CD59*, *LMNA*, *FOS*, *CCL2*, *MUSTN1*, *ZNF638*). Pathway enrichment analysis revealed the activation of pathways involved in glucose and lipid metabolism, such as the PPAR signaling pathway, and related to cholesterol metabolism, glycolysis/gluconeogenesis, fatty acid degradation, and maturity-onset diabetes of the young. These findings confirmed the results of a previous microarray study. Regarding the found miRNAs, eight of them (miR-133B, -210, -22, -25, -27A, -27B, -30E, -503) were associated with obesity, inflammation, T2D, and its microvascular complications, such as diabetic nephropathy and retinopathy [40].

Plasma free fatty acids are known as one of the causes of peripheral insulin resistance [72]. The RNA sequencing of palmitate-treated pancreatic cells of human donors without diabetes demonstrated an expression of 86% (56) of 69 preselected T2D-associated candidate genes, according to previous GWAS. Genes whose expression level was modified by palmitate exposure acted as regulators of cell death and morphology, cellular movement and development, gene expression, molecular transport, lipid metabolism, ubiquitin, and proteasome function. An important finding of the study is the inhibition by palmitate of the transcription factors *PDX1*, *PAX4*, *MAFA*, and *MAFB*, which play a key role in the development of insulin-producing β cells [73].

The combined analysis of gene expression and lipidomics in T2D patients is especially important for understanding the complex relationship between the lipid profile and the development of T2D. The results of integrated lipidomics and transcriptomic analysis revealed a correlation between the altered lipid profile and significantly enriched pathways in the peripheral blood of T2D individuals. The most significant pathways were one carbon pool by folate, arachidonic acid metabolism, and the insulin signaling pathway [19]. Since it is sometimes difficult to draw a direct connection between research findings and T2D and to assess the significance of the found association, comprehensive studies of the kind will allow for the identification of non-obvious relationships which could go unnoticed during isolated gene expression profiling.

### 3.2. DEGs Belonging to Ubiquitin–Proteasome System

A gene expression analysis in pancreatic islets cells using DNA microarrays, performed by Bugliani and colleagues, demonstrated the differential expression of 1285 genes enriched in 23 pathways and numerous functions. Among them, the ubiquitin–proteasome system was consistently affected in the islets of diabetic individuals compared to the controls. A total of 30 genes encoding ubiquitin-conjugating and ubiquitin-ligating enzymes were downregulated, and 4 upregulated genes belonged to the E3 (ubiquitin-ligating) family. Additionally, the expression of *UCHL1*, *USP 1*, *16*, *46*, and *COPS 2*, *3*, and 5, responsible for deubiquitinating enzymes, was downregulated. The differential expression of *UBE2K*, *UCHL1*, and *PSMB7* was confirmed by RT-qPCR. Subsequent immunohistochemistry demonstrated a significantly higher number of β-cells positive for ubiquitination and lower proteasome activity in T2D islets, thus confirming previous findings. The researchers suggested that reduced proteasome activity may impact insulin secretion by islet cells [20]. The expression of genes of the ubiquitin–proteasome system (*UBE2H*, *UBE3A*, *USP2*, *USP54*, *USP30*) was also inhibited in palmitate-treated pancreatic cells, which supports the theory of Bugliani and colleagues regarding the significant influence of proteasome activity on insulin secretion by β cells [73].

### 3.3. DEGs Involved in Immune Response

Corbi and colleagues described DEGs implicated in cell-to-cell signaling and interactions, cellular movement, immune cell trafficking, inflammatory response, and other networks in circulating the lymphocytes and monocytes of T2D individuals with dyslipidemia and periodontitis. Moreover, the DEGs for patients with poorly or well-controlled glycemia were different, which may be of interest for the targeted therapy of T2D. The study once again demonstrated the importance of inflammation in the development of T2D [28]. Using the association rule mining method, the authors analyzed microarray data and identified *CDC42SE2* as a DEG in patients with decompensated T2D, dyslipidemia, and periodontitis—chronic inflammatory diseases [29]. An increased expression of RELA, the p65 subunit of the NF-kappa-B complex which regulates DNA transcription, cell survival, and immune response to infection, were also revealed in gingival tissue affected by periodontitis in T2D patients [74]. The activation of components of the NF-kB signaling pathway was also noted in a study by Saxena et al. [27] and by Lv and colleagues [37]. In an RNA-seq study performed by Lv et al., *LRRC19*, activating NF-kB and stimulating pro-inflammatory cytokines synthesis, was one of the significant DEGs involved in the pathogenesis of T2D.

Another study demonstrated 76 overexpressed genes and 222 suppressed genes common in Han and Kazak Chinese diabetic patients compared to healthy controls. Downregulated genes were mostly enriched in immune response processes such as granulocytes and T-lymphocytes activation and involved in antigen processing pathways. The upregulated genes were associated with fatty acid and carbon metabolism. It seems interesting that, despite the presence of many common networks, the genetic patterns differ in the Han and Kazak Chinese patients, which allows for conclusions to be drawn about the differences in the gene expression in different populations [26]. The overexpression of the *HLA-DQA1* and *HLA-DQB1* genes encoding subunits of MHC class II molecules, which are responsible for antigen presentation by immune cells, was demonstrated in T2D individuals, proving the commonality of immune response activation in T1D, T2D, and latent autoimmune diabetes in adults. The activation of genes associated with the IL-5, IL-6, and IL-17 signaling pathways, as well as with T-cell immune response in patients with well-controlled T2D when compared with poorly controlled T2D, may indicate a better regulation of the immune response in patients with well-controlled T2D [23]. On the other hand, such expression changes can be considered as a potential prognostic marker for the assessment of the effectiveness of hypoglycemic therapy in the future.

Neutrophils are known to play an important role in the development of immune inflammation. In the analysis of the expression profile in neutrophils from T2D patients and healthy controls, 1990 upregulated and 1314 downregulated DEGs were found. The upregulated DEGs were mainly involved in myeloid leukocyte activation, T-cell activation, the adaptive immune system, cytokine production, the immune response-regulating signaling pathway, the cytokine-mediated signaling pathway, immunoregulatory interactions between a lymphoid and a nonlymphoid cell, and the immune response–regulation pathway. The downregulated genes were related to inflammatory response, interleukin-10 signaling, the regulation of cytokine production, and cytokine–cytokine receptor interaction [38].

To shed light on the mechanisms of neutrophil involvement in the inflammatory process in T2D, Kleinstein and co-authors also performed RNA-seq transcriptomic analysis. Having compared the expression profiles of neutrophils in 11 diabetic patients and 7 healthy controls, the authors identified 50 DEGs. The *SLC9A4*, *NECTIN2*, and *PLPP3* genes with the most altered expression in T2D patients were significantly downregulated. Most of the discovered genes are involved in the inflammatory process, as well as in lipid metabolism. The expression of the *GNPDA1* gene was shown to be associated with glucose metabolism [41].

A study performed by Grayson and colleagues discovered the dysregulation of genes associated with T-cell-activation and signaling in T2D individuals compared with healthy controls, once again providing evidence of the involvement of the immune system and cell-mediated immunity in the pathogenesis of T2D [17].

In an analysis of skeletal muscle expression profiles in 271 individuals with glucose tolerances ranging from normal glucose metabolism to clinical manifestation of T2D, the UBN1 (ubinuclein 1) gene was the most differentially expressed [33]. To our knowledge, this gene has not previously been described as associated with T2D or glucose metabolism.

Coronary artery disease (CAD) is a common comorbidity in T2D and is one of the leading causes of death in diabetic patients worldwide [75,76]. In this regard, the results of studies aimed at searching for common expression patterns in individuals with T2D and CAD are of particular interest. According to Gong and colleagues, T2D and CAD patients indeed show similar features of expression profiles in PBMC. Among the overlapping DEGs, genes enriched in terms of viral infectious cycle, anti-apoptosis, endocrine pancreas development, blood coagulation, and innate immune response were the most significant. When comparing the T2D group of patients with healthy controls, the most significant genes were those involved in the G1/S transition of the mitotic cell cycle, blood coagulation, inflammatory response, endocrine pancreas development, and viral reproduction [34]. Previously, researchers also identified common DEGs in CAD and T2D. These included *IL2RA*, *TNFRSF1A*, and *MAPK11* [17], involved in a huge number of cellular processes such as proliferation, differentiation, and transcription regulation. These data once again prove the significant role of inflammation in the pathogenesis of T2D and indicate a complex interplay in the development of T2D and CAD.

The endothelial dysfunction, inevitably leading to atherosclerosis, CAD, and microvascular complications, is known to often accompany T2D [77]. The comparative analysis of gene expression in the endothelial cells of T2D individuals and healthy controls revealed 51 upregulated and 101 downregulated DEGs associated with metabolism and growth pathways [36]. It is evident that the analysis of endothelial cells is not possible in routine practice to identify risk groups for cardiovascular complications in T2D patients. However, this study straightens out the mechanisms involved in endothelial dysfunction and the development of macro- and microvascular complications in T2D.

### 3.4. DEGs Participating in Cancer Signaling and Cell-Cycle Pathways

Also of interest is the upregulation of the cancer signaling pathway, which was observed repeatedly in patients with T2D. For instance, differentially expressed genes in T2D patients and healthy controls were analyzed using microarray technology in the Pakistani population. A total of 1017 genes were downregulated and 640 were upregulated when comparing these two groups. However, the small sample size in this study should be considered (two representatives from each study group). Among the differentially expressed genes identified by the pilot study were those involved in the molecular mechanisms of cancer, cell cycle, DNA replication, recombination and repair, cellular growth and proliferation, cell-to-cell signaling and interaction, and others. Canonical pathway analysis revealed important signaling pathways: insulin receptor signaling, apoptosis signaling, p53 signaling, molecular mechanism of cancer, and cell cycle regulation pathways. The suppression of the p53 tumor suppressor protein gene and the dysregulation of protooncogene MYC in diabetic patients clearly support the theory of an association between T2D and cancer incidence [30]. The data obtained appear consistent with results of other studies on the subject. For instance, the cancer signaling pathway was shown to be the most upregulated pathway in elderly individuals with T2D compared to older people without T2D. In addition, this study also confirmed immune inflammation in T2D, showing the upregulation of T-cell activation and migration and inflammation pathways [31]. These findings reaffirm existing evidence of an increased risk of cancer among T2D patients due to hyperinsulinemia, hyperglycemia, or chronic inflammation [78].

The differential expression of cell cycle genes in the neurons, astrocytes, endothelial cells, and blood cells of diabetic patients was demonstrated by several studies [30,32,36]. The microarray study conducted by Taneera and co-authors was devoted directly to the analysis of the expression profile of cell cycle genes in the islet cells of T2D individuals and healthy controls. The authors reported altered gene expression for cyclins, a cyclin-dependent kinase, and a cyclin-dependent kinase inhibitor. The expression of *CCND1*, *CDK18*, and *CDKN1A* was upregulated, while *CCND3* and *CDK5* were downregulated in T2D patients compared to controls. Moreover, the *CCND1* and *CDK1A* expression levels were shown to correlate with HbA1C levels. *CCND3* expression correlated with both HbA1C and insulin secretion. The expression levels of *CDK5* and *CDK18* positively and negatively correlated with insulin secretion, respectively. An interesting finding of the study was the effect of age on the expression of the *CDKN1A*, *CDKN2A*, *CCNI2*, *CDK3*, and *CDK16* genes [22].

### 3.5. DEGs in T2D Complications

A better understanding of the gene expression changes’ role in diabetic complications might provide novel insights into the genome function mechanisms in T2D, making it possible to define the risk groups for a severe course of disease and to prevent its microvascular and macrovascular complications. The *CD44* and *CCL5* genes were found to be associated with wound healing in diabetic patients when analyzing the transcriptome of dermal lymphatic endothelial cells [79]. *CCL5* is a chemokine that is a chemoattractant for inflammatory cells that release pro-angiogenic factors, stimulating wound vascularization [80]. The transmembrane protein CD44 is also responsible for wound healing, angiogenesis, and immune modulation [81]. Previously published data on the increased concentration of CD44 in the serum of patients with insulin resistance provide hope for the possibility of using this molecule as a potential predictive marker for the development of diabetic complications [82].

Wu and colleagues compared the expression profiles in the skin of 74 T2D patients and 148 healthy controls. The results indicate the involvement of genes responsible for the immune response (*CCL20*, *CXCL9*, *CXCL10*, *CXCL11*, *CXCL13*, *CCL18*), as well as the tumor necrosis factor superfamily and infectious disease pathway genes. The data obtained from RNA-seq correlated with the results from microarray datasets processed in the same way [35]. The overexpression of genes associated with immune response and inflammation in the skin of T2D individuals is consistent with the idea of immune inflammation underlying T2D skin manifestations [83].

The differential expression of 320 genes was found while performing transcriptional profiling comparing the blood-derived RNA of Mexican patients with diabetic retinopathy and T2D individuals without this complication. Confirmation by RT-PCR and cDNA sequencing revealed differential splicing in the *TUBD1* gene encoding Tubulin delta 1. The co-expression of the a and b transcript isoforms was shown to be protective against diabetic retinopathy in T2D patients. The co-expression of the isoforms a, b, and d increased the risk of developing non-proliferative retinopathy, and the absence of the co-expression of the a and b isoforms was associated with the risk of proliferative retinopathy. A conclusion was made about the possible use of *TUBD1* gene transcriptional isoforms as a biomarker for the presymptomatic diagnosis of diabetic retinopathy [25].

To reveal the pathogenesis of neuronal and endothelial damage in T2D, which potentially leads to diabetic neuropathy and dementia, Bury et al. conducted a study on cortical neurons, astrocytes, and endothelial cells, in which 912, 2202, and 1227 genes were shown to be differentially expressed, respectively, when comparing diabetic and nondiabetic patients. The expression level of the selected genes was confirmed using the NanoString nCounter platform. The most significant among them were *DDIT3*, *H2AFX*, *HMOX1*, *NQO1*, *COX5B*, *GRIN1*, and *TF*. The expression and localization of proteins encoded by the *COX5b*, *NDUFb6*, *TGFβ1*, *FOXO3a*, and *p53* genes in neurons were validated using immunohistochemistry analysis. The affected signaling pathways included the HIF-1 signaling pathway, the insulin signaling pathway, cell cycle, cellular senescence, and oxidative phosphorylation signaling. The activation of these neuronal signaling pathways can be associated with neuronal dysfunction in dementia, and changes in the gene expression in astrocytes and endothelial cells indicate that T2D also impacts non-neuronal cells in the central nervous system [32].

### 3.6. The Role of Ethnic Differences

Of particular interest are studies on the gene expression in different ethnicities representatives, since it is known that African American origin is a risk factor of T2D development [84]. The results of a comparative analysis of the expression profiles in the white blood cells of African American and Caucasian origin subjects demonstrated statistically significant differences. The communication between innate and adaptive immune cells and primary immunodeficiency signaling pathways was significantly downregulated in African American subjects, while the complement system and interferon signaling pathways were downregulated in individuals of Caucasian origin. The researchers also found common patterns specific to T2D patients. Thirty DEGs were detected when compared with the control group. The expression of *ABCA1*, *ALAS2*, *IL4*, *IL8*, and *PTGS2* was validated by qPCR. The identified DEGs were implicated in immune responses, lipid metabolism, and organismal injury/abnormality [18].

### 3.7. Early Expression Changes

To identify the specific expression patterns of insulin resistance, a transcriptomic profile of peripheral blood mononuclear cells was analyzed in healthy subjects with extremes of the distribution of HOMA-IR. A total of nine genes from Adrenergic Signaling in the CardioMyocytes pathway with a significantly altered expression in PBMC (*ADCY9*, *TNNI3*, *RAPGEF3*, *CACNA1S*, *CACNG3*, *ADRA1B*, *CAMK2D*, *PPP2R3C*, *PPP2CA*) were identified when comparing subjects with extreme insulin resistance or sensitivity. The function of these genes is to regulate calcium influx, cell growth, and apoptosis [24]. Even though these genes are not directly involved in the regulation of glucose homeostasis, this finding in healthy and young individuals provides an opportunity for the further search for genetic markers of the risk of developing T2D.

The search for expression changes common for T2D and impaired fasting glucose brings us closer to universal biomarkers that allow for the identification of risk groups for developing T2D. These biomarkers can include both DEGs and circulating miRNAs. Cui and co-authors found the overexpressed *TAF1* gene and downregulated *MAFK* as common nodes in T2D individuals and subjects with impaired fasting glucose [85]. The *TAF1* gene was associated with insulin resistance and glycated hemoglobin and was overexpressed in obese T2D subjects in previous studies [86,87]. The *MAFK* gene may also play an important role in T2D pathogenesis, as it regulates numerous β-cell-related genes [88]. The researchers also found common dysregulated miRNAs: miR-29a, miR-144, and miR-192 [85]. The dysregulation of these microRNAs has already been described in previous studies comparing T2D subjects and healthy controls [89].

### 3.8. MicroRNA Expression Changes

Over the past decade, researchers repeatedly demonstrated miRNA expression changes in T2D. The availability of circulating miRNAs makes them promising biomarkers for the presymptomatic diagnosis of T2D. The expression profile changes of multiple circulating miRNAs were demonstrated in studies on diabetic patients [89,90]. MiR-29a, miR-144, miR-375, and miR-34a are among the most studied miRNAs in T2D patients. MiR-375 is highly expressed in pancreatic islets [91], whole blood [92], plasma [93], and serum [94], and it is involved in insulin secretion and glucose homeostasis [95]. MiR-34a regulates the inhibition of cell-cycle progression and the induction of apoptosis, associated with p53 function [96]. The expression changes of miR-34a were also detected in the PBMC [97], serum [94], and plasma [98] of diabetic patients. The target gene for miR-29a and miR-144 is *IRS1*, a mediator between the insulin receptor, insulin-like growth factor 1, and PI3K/AKT intracellular signaling pathway elements involved in the translocation of GLUT4 into the plasma membrane for glucose transport [99,100,101]. According to early studies, miR-144 was found in serum [102], plasma [103], muscle [104], and whole blood [92]. MiR-29a, in turn, was expressed in all insulin-dependent organs and tissues [105]. Altered levels of miR-29a were also detected in plasma [106], serum [95], and whole blood [92].

## 4. Common Features of Gene Expression in Type 1 Diabetes and Gestational Diabetes Mellitus

An interesting finding was described by Chinese authors who showed that a decreased expression of jun proto-oncogene (*JUN*) and the upregulation of interleukin 1β (IL-1β) in peripheral blood may have a prognostic value for T2D development in children [107]. Interestingly, the differential expression of the *IL1B* gene has already been described in children with T2D by Kaiser and colleagues. The study analyzed the expression profiles in children with type 1 diabetes (T1D) and T2D. The comparison showed that the *IL1B* gene belongs in common for T1D and T2D DEGs, while the expression of *MYC* is significantly increased in T1D children when compared with T2D children [16]. IL-1β plays a significant role in the pathogenesis of diabetes, as it is known to regulate the inflammatory process, leading to β-cell death [108].

The search for common pathogenetic mechanisms and prognostic markers for T1D and T2D is also of particular interest. Gao and colleagues compared the T1D and T2D expression profiles with healthy controls. The *MGAM* and *NAMPT* genes had a high predictive value for T2D. *PNP* and *CCR1* were common DEGs for T2D and T1D [109]. To date, there are no studies to explain the relationship between the *PNP* gene and diabetes, but it is known that antagonists of *CCR1* (C-C chemokine receptor type 1) can inhibit the recruitment of monocytes and lymphocytes into the renal interstitium and thus have a beneficial effect in slowing down the development of diabetic nephropathy [110]. The suppression of *CCR1* also attenuates neuropathic pain in diabetic mice [111]. *MGAM* (maltase-glucoamylase) is known to modulate gluconeogenesis and has a high hydrolytic activity against glucose oligomers [112,113]. The *NAMPT* gene was also repeatedly reported as associated with T2D and obesity [114].

In another microarray study, the expression profiles in the peripheral mononuclear blood cells of patients with T1D were closer to gestational diabetes mellitus (GDM) as compared to T2D. The DEGs common to T1D, T2D, and GDM were 22 genes, 7 of which (*EGF*, *FAM46C*, *HBEGF*, *ID1*, *SH3BGRL2*, *VEPH1*, and *TMEM158*) proved to be the most significant [21]. The epidermal growth factor (EGF) was shown to regulate insulin secretion, and its level was reduced in diabetic mice [115]. *FAM46C*, being a target gene for miR-657, is involved in the pathogenesis of GDM by regulating macrophage proliferation and migration and thus modulating the inflammatory response in GDM [116]. The heparin-binding EGF-like growth factor encoded by the *HBEGF* gene is crucial for glucose-stimulated β-cell proliferation, as demonstrated in rats [117]. The DNA-binding protein inhibitor ID1 was found in the islets of diabetic mice and is also associated with diabetes. ID1 knockout plays a protective role and prevents the development of glucose intolerance and the loss of b-cell gene expression induced by a high-fat diet [118]. Rs9352745, the expression quantitative trait locus for *SH3BGRL2* gene, is associated with gestational weight gain [119]. This may partly explain the excess weight gain during pregnancy in patients with GDM. The role of the *VEPH1* gene in the development of diabetes has not yet been studied, but Melted, an ortholog of *VEPH1* in Drosophila, is known to be involved in insulin/PI3K signaling [120]. The role of the last gene, *TMEM158*, in the pathogenesis of T2D has yet to be studied. Despite the lack of a comprehensive theory that fully explains the significance of the aforementioned genes in the development of T2D, T1D, and GDM, the search for common mechanisms can contribute to the improvement of approaches to the prevention and precision treatment of diabetes.

## 5. Findings from Single-Cell Sequencing Studies

Great knowledge about the processes undergone directly in the pancreas was obtained from single cell studies. Using this new technology, researchers have shown the high proliferative capacity of multipotent pancreatic progenitor cells, as well as their developmental heterogeneity and dedifferentiation processes in adult T2D individuals. Segerstolpe et al. applied another approach to the transcriptome analysis of pancreatic islet cells. The single-cell transcriptome profiling of endocrine and exocrine pancreatic cells obtained from six healthy controls and four T2D donors revealed the downregulated expression of *INS* and *FXYD2* genes and the overexpression of *GPD2* and *LEPROTL1* in the β-cells of T2D individuals. All of the genes were previously reported to be involved in T2D pathogenesis through modulating insulin secretion and regulating body weight. The expression of *PCSK1N* and *SCG5* was found to correlate with the body mass index in all cell types of the pancreatic islets [42].

A similar conclusion was reached by Lawlor and colleagues, who also found a decreased expression of *INS* along with *STX1A* in T2D β-cells compared with nondiabetic donors. Both detected genes play a significant role in insulin secretion by β-cells. Among other things, the researchers also found an upregulated expression of CD36 and a downregulated expression of GDA in α-cells. CD36, a fatty acid transporter, was previously shown to be associated with diabetes, while the role of *LAPTM4B* and *RCOR1*, significantly dysregulated in δ-cells, remains unclear and requires further investigation [45,121].

Another study based on a single-cell RNA-seq of islet cells from 6 diabetic and 12 non-diabetic organ donors revealed the dysregulated expression of 54, 48, 119, and 33 genes in α-, β-, and δ-cells and PP cells, respectively. Most of them have an unknown function (28%) or are related to non-islet cell growth (38%) or another non-islet cell function (26%) [43].

A study comparing the α- and β-cell expression profiles of T2D donors and children showed intriguing results [44]. One of the main findings of the study reveals that the islet cells’ expression profile in child donors had immature gene signatures. Indeed, islet cell differentiation is known to complete after maturation in the postnatal period [122]. The expression profiles in T2D donors were partially like those in children, demonstrating the dedifferentiation process that causes the dysfunction of insulin secretion, which may be one of the key mechanisms in the development of T2D, along with insulin resistance. The dedifferentiation of β-cells into a multipotent progenitor state may be caused by hyperglycemia, as previously shown (glucose toxicity) [123]. However, the expression profiles of β-cells in healthy old and young mice are very similar, which may indicate that the processes of β-cell dedifferentiation occur not only under the influence of hyperglycemia but also in the normal state [124].

According to the results of another single-cell study on islet cells from mice, the differentiation and increase of β-cells are regulated by reactive oxygen species and Srf/Jun/Fos transcription factors [125]. At the same time, heterogeneity in the regulation of genes responsible for functional maturation and endoplasmic reticulum (ER) stress was also found in the β-cell population [126]. It was shown that the loss of functional b-cell mass due to the dedifferentiation of β-cells and their loss of maturity might be in response to ER stress [127]. Since recent studies have shown the reversibility of the dedifferentiation process, this gives hope that suppressing ER stress will help to restore lost functions.

Interestingly, a single-cell study performed on isolated β-cells from healthy donors also showed islet cell dedifferentiation and transdifferentiation. In particular, insulin-secreting cells showed a low expression level of β-cell genes and an overexpression of progenitor markers. β-cells were assumed to be in a plastic state and undergo dedifferentiation during isolation and ex vivo cultivation [128]. Additionally, researchers showed an independent accumulation of genetic aberrations and transcriptional noise in islet cells with age [129]. Undoubtedly, these age-related changes can affect, among other things, the endocrine function of the pancreas, which can partly explain the development of T2D in middle-aged and older adults.

Later, in another single-cell study, pancreatic cells (in particular, progenitor ductal cells) from human donors demonstrated developmental heterogeneity [130]. An important conclusion of this work is that, in addition to exocrine and endocrine progenitors, there may be a ducto-endocrine differentiation gradient. In addition, these cells were found in the pancreas of T1D and T2D donors regardless of disease duration. The possibility of stimulating these cells in vivo opens new opportunities for further pharmacological research and the elaboration of novel therapeutic approaches. The results of recent research put before us the forgotten concept of the plastic state of ductal progenitor cells capable of differentiation into facultative progenitors and dedifferentiation [131].

Single-cell transcriptomic studies on islet cells have become one of the most popular molecular methods in the study of T2D in the past years, since it requires a relatively small amount of material and allows for the determination of the expression signatures of individual cells, including those with a low abundance. One of the main difficulties in conducting a single-cell experiment is the choice of sample preparation protocol. Bonnycastle and colleagues demonstrated the importance of proper fixation and the cryopreservation of islet cell samples for further single-cell sequencing. The researchers found significant differences in the proportion of the detected cell types, as well as in the specific patterns of gene expression in the islets prepared for a single cell experiment using different protocols [132]. This indicates the need to develop a standardized approach to sample preparation methods, sequencing, and bioinformatics analysis in the single-cell study by investigators. A small number of the analyzed samples of islet cells derived from donors, due to difficulties in obtaining material for analysis, may manifest in the absence of the replication of research results. For instance, Wang and colleagues analyzed the results of three previously published single-cell studies in T2D patients and control groups and found a limited number of overlapping genes expressed by β-cells. These include downregulated *ERO1LB* and upregulated *TTR*, *FXYD2*, *TUBA1C*, *FTL*, and *INS* [133]. Another issue of current interest is the limited number of reliable α- and β-cell markers, which would help to separate cell populations. The use of flow cytometry made it possible to determine the marker TM4SF4 specific for α-cells, as well as CD24 and CD44 localized on the membranes of acinar and ductal cells, while β-cells turned out to be TM4SF4- and CD24-negative [134]. These findings can undoubtedly be extremely useful and beneficial for obtaining data specific to the populations of islet cells. However, the problem of the morphofunctional and molecular heterogeneity of β-cells remains. Protein expression studies came to the rescue in solving this issue, which made it possible to identify surface proteins and intercellular molecules specific for subpopulations of β-cells. Thus, the differential expression of *ST8SIA1* and *CD9* allows for the distinguishing between subpopulations of β-cells in normal adult islets and for the revelation of significant differences in terms of β-cells subpopulation percentages in diabetic donors [135,136]. A deeper understanding of the β-cells population heterogeneity will allow us to develop strategies for the prevention and targeted therapy of T2D in the future.

## 6. Comparative Analysis of Genomic and Transcriptomic Patterns

Of particular interest is the question of how the expression changes obtained from microarrays and RNA-seq studies correspond to molecular pathways and biological processes associated with genes at loci associated with T2D in major GWAS. To answer this question, we performed a comparative analysis of genomic and transcriptomic patterns based on the datasets from the open databases. First, we obtained a complete list of 628 genes at GWAS loci using several recent GWAS datasets for T2D, including the UK Biobank study, GERA [137], and latest meta-analyses involving the DIAGRAM consortium [138,139]. The analysis was performed using a recently proposed method for the functional annotation of GWAS data called FUMA [140]. We also selected 2492 DEGs with the most significant log2-feld change obtained in studies from Table 1. The gene list provided by FUMA was intersected with a complete list of DEGs. Such an analysis identifies 107 common genes between the two sets (Figure 2a). We next analyzed the molecular pathways associated with these common genes. Enrichment analysis using the Molecular Signatures Database [141] identified 98 canonical pathways, showing a significant overlap with 107 common genes. The top enrichment included processes related to the cell cycle, mitosis, and immune system activity (Figure 2b). The relatively small number of shared genes may be due to the existing limitations of both transcriptomic and GWA studies. In addition, the analysis of ethnically heterogeneous populations and different tissues may mask existing differences. At the same time, the recent rapid growth in the number of transcriptomic studies gives hope for additional data allowing for complex meta-analyses with strict inclusion and exclusion criteria.

## 7. Discussion

To date, the contribution of the hereditary component to the development of T2D is well known; however, the pathogenesis of this disease is not yet fully understood. The multifactorial nature of T2D makes it challenging to study, so it seems promising to identify the genetic factors that have a significant impact on the development of the disease throughout the application of omics approaches, including genomics, transcriptomics, proteomics, and others. Over the past decade, our knowledge on the functional pathways of T2D has been greatly expanded by recent transcriptomic studies that confirmed the important role of the immune system and inflammation in diabetes development as well as indicated a clear involvement of the components of the cancer signaling pathway, which is consistent with the hypothesis that the risk of cancer is significantly higher in T2D patients [78]. The significant role of visceral adipose tissue in the development of insulin resistance in T2D was confirmed by RNA-seq research [39]. Additionally, the common molecular mechanisms in the pathogenesis of T2D, T1D, and GDM were found, as well as early presymptomatic changes in individuals with borderline insulin resistance and impaired fasting glucose [24,85].

Conducting a transcriptomic study requires the consideration of many factors that may affect the results obtained. As is known, T2D patients often have concomitant diseases, an elevated BMI, and diabetic macro- and microvascular complications influencing the expression profile, and this should be considered when forming study groups and planning experiments. In addition, it is necessary to take into consideration hypoglycemic therapy and medications for comorbidities that are taken by patients and their effectiveness. The ethnicity of the participants can also play a significant role. It was suggested that the molecular mechanisms of the T2D pathogenesis may differ in the representatives of diverse ethnic groups [26].

One of the main points during the preparation of a transcriptome analysis experiment is the choice of study material. Since the main insulin targets are the liver, skeletal muscle, and adipose tissue, studying the expression in these tissues really matters [142]. The main difficulty is the need for a biopsy or another invasive procedure so that a piece of tissue or a sample of cells could be tested. Whole blood or PBMC might be a solution to this issue. Peripheral blood testing is more suitable for searching for potential T2D risk biomarkers or for early diagnosis due to its accessibility. Of particular interest are studies on the pancreas; however, obtaining a biomaterial is difficult. The material from deceased donors, cell cultures, or mouse models of T2D may solve this problem. T2D mouse models are currently being used by many researchers around the world [143]. It was shown that the expression profiles of mouse and human α- and β-cells are very similar; however, there are interspecies differences that do not allow for the full characterization of molecular and biological processes in T2D individuals [43] and the prediction of the regenerative behavior of human pancreatic tissue [131]. Additionally, the experiment often requires special controlled conditions. These limitations and ethical considerations limit the use of animal models to study T2D. The use of cell cultures is devoid of the listed shortcomings since it allows the experiment to be carried out under specified conditions. Today, there is a wide variety of in vitro models of pancreatic cells, each of which has its own advantages and disadvantages. Some of them, i.e., primary cell lines, are more representative but have a limited proliferative potential and are difficult to obtain. Transformed cell lines and stem cells, in contrast, have a longer lifespan but are less representative of the selected tissue [144]. Scientists are still searching for the ideal in vitro model of the pancreas for gene expression research.

Due to the high cost and complex sample preparation, the sample size used for transcriptomic studies is commonly quite small. A small sample size is one of the basic reasons for validating study results. In the case of a small sample, it makes sense to validate the expression of selected genes by the RT-qPCR method on an expanded sample to obtain reliable results, since the RT-qPCR method is more accessible and easier to perform compared to RNA-seq or expression arrays. One of the core issues of most conducted transcriptomic studies is the lack of validation by another method. Since transcriptomic studies provide massive data on a variety of DEGs, researchers are faced with a choice about the set of genes that require validation. These may be genes with the most significant expression changes or genes with a described function. When comparing the data on gene expression obtained by RNA-seq and RT-qPCR, they were reproduced in 85% of cases. However, the remaining 15% consisted mainly of genes with fewer exons or a lower expression. When conducting transcriptomic studies, the validation of the expression levels of such genes by RT-qPCR is required [145].

The interpretation of the study results can also be difficult, since the found DEGs often do not have the described function. Additional studies aimed at the analysis of non-coding RNAs, DNA methylation, and histone acetylation, which can affect the level of expression, as well as proteomic studies, may partly bring us closer to solving the problem by determining genes that are differentially expressed at both the transcript and protein levels and factors affecting expression.

## 8. Conclusions

In this article, we present the main results of transcriptomic studies conducted using expression arrays, RNA-seq, and single cell sequencing. Recent research provided evidence for the important role of immune inflammation in the pathogenesis of T2D and revealed the activation of cancer signaling, cell cycle pathways, and the role of the ubiquitin–proteasome system in T2D patients. The importance of adipose tissue, both visceral and peripheral, in the development of insulin resistance was confirmed once again. Despite the massive data provided by transcriptome studies, there are still many limitations affecting the results, as well as challenges that researchers must face from the choice of the material and method to the interpretation of the results. Undoubtedly, the data obtained using transcriptomic technologies allow for the shedding of light on the molecular mechanisms of T2D pathogenesis and the consideration of new diagnostic and therapeutic approaches. However, before introducing them into clinical practice, additional large-scale studies are required, considering both the achievements and the limitations of previous studies.

## Figures and Tables

**Figure 1 genes-13-01176-f001:**
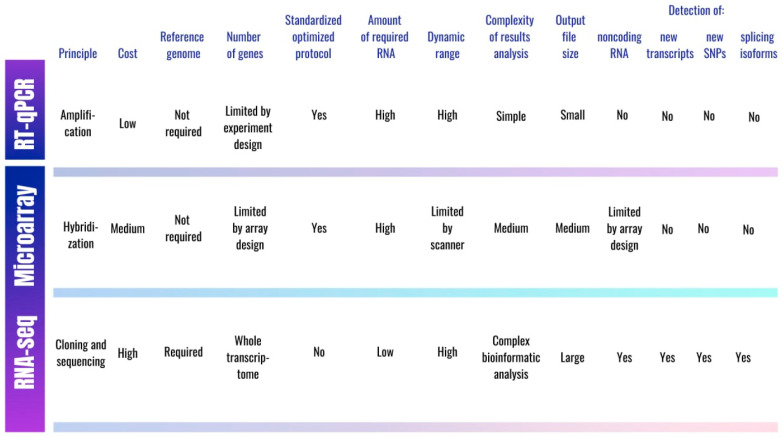
Comparison of approaches to gene expression analysis.

**Figure 2 genes-13-01176-f002:**
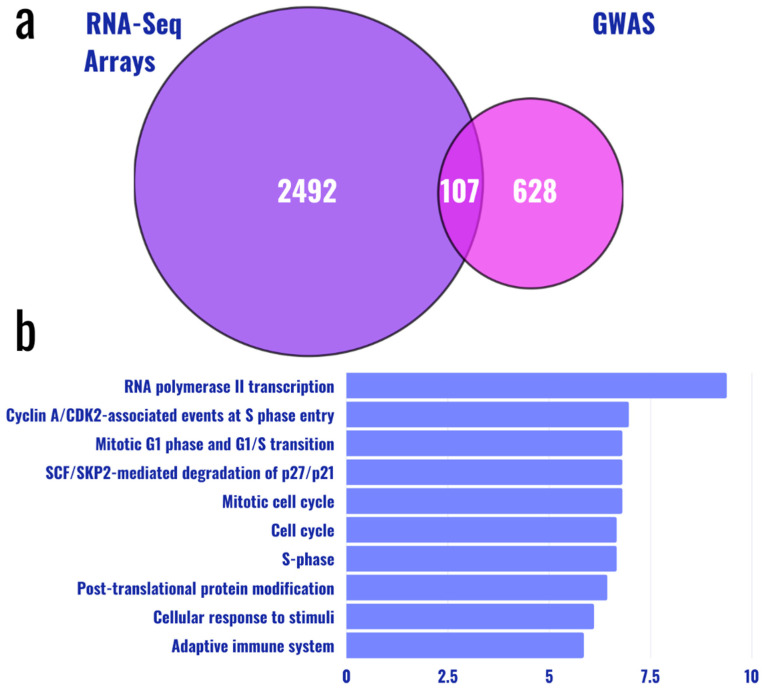
(**a**) Venn diagram showing the overlap between genes at significant GWAS loci for T2D and DEGs identified using RNA-seq and arrays in T2D patients; (**b**) A bar plot showing false discovery rate-adjusted enrichment *p*-values for the top 10 gene sets with a significant enrichment among 107 T2D DEGs identified by both transcriptomic studies and GWAS.

**Table 1 genes-13-01176-t001:** Transcriptomic studies in T2D patients.

Sample Type	Transcriptomic Technique	Validation	Study Group	References
PBMC	Microarray	qPCR	43 newly diagnosed T1D, 12 newly diagnosed T2D, 24 HC	[16]
Whole blood	Microarray	qPCR	6 patients with metabolic syndrome, 6 CAD, 8 T2D, 6 rheumatoid arthritis patients, 9 HC	[17]
Whole blood	Microarray	qPCR	84 T2D, 60 HC	[18,19]
Human islet cells	Microarray	–	7 non-diabetic subjects, 6 T2D donors	[20]
Whole blood	Microarray	–	19 T1D, 20 T2D, 17 GDM	[21]
Human islet cells	Microarray	qPCR	67 non-diabetic donors, 10 T2D donors	[22]
PBMC	Microarray	qPCR	5 poorly controlled T2D, 7 well-controlled T2D, 6 normoglycemic individuals	[23]
PBMC	Microarray	–	10 healthy individuals with extreme insulin resistance, 10 healthy individuals with extreme insulin sensitivity	[24]
Whole blood	Microarray	qPCR	20 T2D with diabetic retinopathy, 10 T2D without diabetic retinopathy	[25]
Abdominal omental adipose tissues	Microarray	–	12 T2D, 12 HC	[26]
Adipose tissue from thigh	Microarray	qPCR	30 T2D, 30 HC	[27]
PBMC	Microarray	qPCR	5 poorly controlled T2D with dyslipidemia and periodontitis, 7 well-controlled T2D with dyslipidemia and periodontitis, 6 normoglycemic with dyslipidemia and periodontitis, 6 healthy individuals with periodontitis, 6 HC	[28,29]
Whole blood	Microarray	TaqMan Low Density Array	2 T2D, 2 HC	[30]
Whole blood	Microarray	–	12 T2D, 19 HC	[31]
Neurone, astrocyte, and endothelial cell	Microarray	NanoString nCounter platform + Immunohistochemical validation of protein expression	6 T2D, 6 HC	[32]
Skeletal muscle	RNA-Seq	–	271 participants with glucose tolerance ranging from normal to newly diagnosed T2D	[33]
PBMC	RNA-Seq	–	2 T2D, 2 CAD, 6 T2D + CAD, 7 HC	[34]
Skin samples	RNA-Seq	–	74 T2D, 148 HC	[35]
Endothelial cells from cubital vein	RNA-Seq	–	5 T2D, 5 HC	[36]
Whole blood	RNA-Seq	–	6 T2D with thirst and fatigue, 6 HC	[37]
Neutrophils	RNA-Seq	qPCR	5 newly diagnosed T2D, 5 HC	[38]
Visceral adipose tissue	RNA-Seq	qPCR	10 T2D, 10 HC	[39]
Adipose tissue from thigh	RNA-Seq	qPCR	5 T2D, 5 HC	[40]
Neutrophils	RNA-Seq	–	11 T2D, 7 HC	[41]
Human islet cells	Single-cell RNA-Seq	RNA in situ hybridization	4 T2D, 6 HC	[42]
Human islet cells	Single-cell RNA-Seq	–	6 T2D, 12 HC	[43]
Human islet cells	Single-cell RNA-Seq	–	1 T1D donor, 3 T2D donors, 2 children, 3 HC	[44]
Human islet cells	Single-cell RNA-Seq	RNA in situ hybridization	3 T2D, 5 HC	[45]

PBMC—peripheral blood mononuclear cells; HC—healthy controls; T1D—type 1 diabetes; T2D—type 2 diabetes mellitus; GDM—gestational diabetes mellitus; CAD—coronary artery disease. Studies are divided by method (microarray, RNA-seq, single-cell RNA-Seq).

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
