# Peer review of "Overview of Transcriptomic Research on Type 2 Diabetes: Challenges and Perspectives"

_genes, 2022, doi:10.3390/genes13071176_

Round 1

Reviewer 1 Report

This manuscript summarized the progress of research using transcriptomics to study type 2 diabetes and provides a more comprehensive summary of commonly used transcriptomics research techniques and studies of diseases associated with type 2 diabetes. The summary is interesting and provides a reference for the clinical management of type 2 diabetes, but there are some minor issues, as follows:

1 At the beginning of Part 3 (Transcriptome studies in T2D), it is suggested that a paragraph be added to provide a brief description of the relevance of type 2 diabetes to the content of the summary, such as type 2 diabetes and lipid metabolism, the ubiquitin-proteasome system, and the immune system, which are also not mentioned in the introduction

2 The following 11 subheadings in Part 3 seem a bit too many, they are not exactly parallel, it seems that 3.1-3.6 are all studies directly related to type 2 diabetes, it is suggested that the content of the 11 subheadings in Part 3 be described in categories

3 The summary is comprehensive, but it seems to be missing the most critical summary (table) of gene expression related to type 2 diabetes, which is important for a review

4 The images in the text are not sufficiently clear and it is recommended to adjust them

Reviewer 2 Report

The authors presented a very nice overview of recent progress in type 2 diabetes (T2D) using transcriptomic techniques. This review summarized the evolvements of transcriptomic studies in T2D and some major findings associated with the pathological mechanisms of the disease. The overall writing is coherent and smooth.

I have one moderate and a few minor comments for this manuscript.

1.     The differentially expressed genes (DEGs) have been a crucial part in this review, especially in section “3. Transcriptome studies in T2D”, but the authors did not introduce how DEGs are defined or detected. I would suggest the authors add a bit more details about the algorithms and methods for DEG analysis. The authors could refer to the review paper “A survey of best practices for RNA-seq data analysis”, which has been cited >2000 times, for more information.

2.     In Table 1, the titles should be specific, e.g., sample type, transcriptomic techniques, etc.

3.     In Abstract, line 20, “has the potential to reveal significant insights into”, reveal should be provide.

4.     In Figure 1, “Dinamic range” should be “Dynamic range”.

Author Response

The authors presented a very nice overview of recent progress in type 2 diabetes (T2D) using transcriptomic techniques. This review summarized the evolvements of transcriptomic studies in T2D and some major findings associated with the pathological mechanisms of the disease. The overall writing is coherent and smooth. I have one moderate and a few minor comments for this manuscript.

Response: We thank the Reviewer for the positive assessment of our work as well as for the useful comments and suggestions.

Point 1. The differentially expressed genes (DEGs) have been a crucial part in this review, especially in section “3. Transcriptome studies in T2D”, but the authors did not introduce how DEGs are defined or detected. I would suggest the authors add a bit more details about the algorithms and methods for DEG analysis. The authors could refer to the review paper “A survey of best practices for RNA-seq data analysis”, which has been cited >2000 times, for more information.

Response 1: We thank the Reviewer for the valuable suggestion. The section “3. Transcriptome studies in T2D” was updated. The information about principles and approaches to DEG analysis was added. 

Point 2. In Table 1, the titles should be specific, e.g., sample type, transcriptomic techniques, etc.

Response 2: We specified the column titles in Table 1.

Point 3. In Abstract, line 20, “has the potential to reveal significant insights into”, reveal should be provide. 

Response 3: We rewrote this sentence and briefly described the potential of transcriptome technologies.

Point 4. In Figure 1, “Dinamic range” should be “Dynamic range”.

Response 4: We replaced “Dinamic range” with “Dynamic range” in Figure 1.